# Impact of Host Telomere Length on HHV-6 Integration

**DOI:** 10.3390/v14091864

**Published:** 2022-08-24

**Authors:** Darren J. Wight, Giulia Aimola, Georg Beythien, Louis Flamand, Benedikt B. Kaufer

**Affiliations:** 1Institut für Virologie, Freie Universität Berlin, Robert von Ostertag-Straße 7-13, 14163 Berlin, Germany; 2Division of Infectious and Immune Diseases, CHU de Quebec Research Center-Laval University, Québec, QC G1V 4G2, Canada; 3Department of Microbiology, Infectious Disease and Immunology, Faculty of Medicine, Laval University, Québec, QC G1V 4G2, Canada; 4Veterinary Centre for Resistance Research (TZR), Freie Universität Berlin, 14163 Berlin, Germany

**Keywords:** human herpesvirus 6, HHV-6, telomeres, integration, telomere length, TZAP

## Abstract

Human herpesvirus 6A and 6B are two closely related viruses that infect almost all humans. In contrast to most herpesviruses, HHV-6A/B can integrate their genomes into the telomeres during the infection process. Both viruses can also integrate in germ cells and subsequently be inherited in children. How HHV-6A/B integrate into host telomeres and the consequences of this remain a subject of active research. Here, we developed a method to measure telomere length by quantitative fluorescence in situ hybridization, confocal microscopy, and computational processing. This method was validated using a panel of HeLa cells having short or long telomeres. These cell lines were infected with HHV-6A, revealing that the virus could efficiently integrate into telomeres independent of their length. Furthermore, we assessed the telomere lengths after HHV-6A integration and found that the virus-containing telomeres display a variety of lengths, suggesting that either telomere length is restored after integration or telomeres are not shortened by integration. Our results highlight new aspects of HHV-6A/B biology and the role of telomere length on virus integration.

## 1. Introduction

Mammalian telomeres are long stretches of TTAGGG repeats that cap the ends of linear chromosomes [1,2]. Human telomeres range in size from 5 to 15 kb and function as a protective capping structure [3]. The shelterin protein complex assembles on the repeat sequences and effectively hides the chromosome ends from incorrect recognition as dsDNA breaks, leading to unwanted recombination and chromosomal fusion [4,5,6]. Due to the end-replication problem, telomeres shorten with each cell division, until a critical length is reached and the cell either dies or enters senescence [7,8,9,10,11,12]. This makes telomere length a molecular clock, timing the replicative capacity of cells and tissues as we age. To maintain the replicative capacity of stem and progenitor cells, telomere shortening is counterbalanced by the action of telomerase, which adds new telomeric sequences to the ends of chromosomes [13,14,15,16,17]. Importantly, telomere length homeostasis is more complex than this, and different factors are involved in regulating this process; among others, the telomeric zinc finger-associated protein (TZAP), also known as ZBTB48, has been proposed to have a trimming effect on telomere length [18,19].

Human herpesvirus 6A and 6B (HHV-6A/B) are two closely related betaherpesviruses [20,21,22]. Infection of humans with these viruses is widespread, with almost all humans infected with HHV-6B by the age of 2 years, while HHV-6A infection is thought to be acquired later in life [23,24]. As with all herpesviruses, HHV-6A/B establish a life-long persistent infection after primary infection [23,25]. However, in contrast to most herpesviruses, HHV-6A/B are found integrated into the telomeres of latently infected cells [26,27,28,29,30]. This integration can occur not only in many different somatic cell types but also in germ cells. Integrated HHV-6A/B can subsequently be inherited through the germline in a Mendelian fashion [25,26,31,32]. These individuals with inherited chromosomally integrated HHV-6 (iciHHV-6) represent ~1% of the human population, with slight prevalence variations depending on the geographical location. Little is known about the consequences of carrying a large integrated virus in a human telomere, although iciHHV-6 patients are over-represented in hospital admissions and have increased risk of angina pectoris and pre-eclampsia, suggesting there may be an important impact on human health [22,33,34,35,36,37,38,39].

How HHV-6A/B are able to integrate into telomeres is not fully understood [40,41]. We know that viral telomeric sequences (TMRs) at the ends of the HHV-6A/B genome that are identical to host telomeres play a crucial role in integration [42,43,44,45]. This has furthered the idea that HHV-6A/B integrate through homology-directed repair between viral TMRs and the host telomere, although a specific recombination pathway has not yet been identified [46,47]. HHV-6A/B encode a putative integrase termed U94 that was considered as a likely candidate for HHV-6A/B integration based on its biological properties; however, U94 was found to be dispensable for integration, at least in vitro [47,48]. We and others have also looked at other factors, both viral and cellular, without establishing a key set of proteins responsible for the integration mechanism, suggesting that HHV-6A/B likely exploit several different pathways to achieve integration [40,46,49,50]. Thus, any information regarding the HHV-6-telomere interaction is vital to provide a better understanding of this process and its biological consequences.

Here, we show for the first time that HHV-6A integration efficiency is not affected by the length of the host telomeres in vitro. Furthermore, we provide evidence that following integration, telomeres of varying length are present, suggesting that telomere integration per se does not lead to robust telomere shortening. Lastly, a newly described telomere-shortening molecule, TZAP, did not influence HHV-6A integration. These observations highlight key aspects of the integration process that were previously undescribed.

## 2. Materials and Methods

### 2.1. Cells and Virus

HeLa Kyoto-based cell lines were maintained in MEM (PAN Biotech; Aidenbach, Germany), 10% fetal bovine serum (FBS; PAN Biotech), 1 mM sodium pyruvate (PAN Biotech), and penicillin/streptomycin. The 293T cells were maintained in DMEM (PAN Biotech), 10% fetal bovine serum (FBS; PAN Biotech), 1 mM sodium pyruvate (PAN Biotech) penicillin/streptomycin, and 5 µg/mL plasmocin (InvivoGen; Toulouse, France). Cells were cultured in a humid incubator with 5% CO_2_. HHV-6A GFP reporter BAC constructed from the U1102 strain was produced as described previously in JJhan cells [42,51].

### 2.2. HHV-6A Integration Assay

The assay was performed as described previously [42]. Briefly, HeLa or 293T cells were cocultured with GFP-reporter HHV-6A (HHV-6A-GFP)-infected JJhans for 4 h. JJhans were then washed away and, the following day, the GFP-positive cells sorted on a FACS AriaIII (BD Biosciences; Franklin Lakes, NJ, USA) or quantified using a CytoFLEX flow cytometer (Beckman Coulter; Brea, CA, USA) for normalization. Immediately post-sorting, a sample was taken for DNA extraction, and the remaining cells were returned to culture. Cells were passaged for a period of time (specified in the text), and a sample was taken for DNA isolation and/or was prepared for fluorescent in situ hybridization, as previously described [42,51,52].

### 2.3. Quantitative PCR

DNA was isolated from samples using an RTP DNA/RNA Virus Mini Kit (Stratec Biomedical; Beringen, Switzerland) following the manufacturer’s guidelines. The qPCR was performed as previously described [42,46].

### 2.4. Telomere Quantitative Fluorescent In Situ Hybridization

Cells were prepared, fixed, and spread as previously described [42,51,52]. A peptide nucleic acid (PNA) probe against the C-strand of telomeres and conjugated with Alexa Fluor 647 (TelC-Alexa647) was used (PNA Bio; Old Conejo Rd, CA, USA). Staining was performed as written in the manufacturer’s protocol. Briefly, slides with cell spreads were treated in pepsin and RNase, then dehydrated and dried. TelC-Alexa647 was heated to 55 °C, then cooled before addition to 85 °C preheated hybridization buffer. We applied 200 nM of probe to the slides, which we sealed and heated to 85 °C for 10 min. We then incubated the cells in the dark for 2 h. Cells were washed twice in wash buffer (2× SSC 0.1% Tween-20) at 60 °C and once in room-temperature wash buffer. Slides were counterstained with DAPI and mounted with Prolong glass mounting media (Invitrogen; Carlsbad, CA, USA).

### 2.5. HHV-6 and Telomere qFISH

FISH was performed as previously described above with the following modifications for TelC-Alexafluor647. We performed treatment with pepsin for 5 min and 2× SSC RNAse A for 4 min. TelC-Alexa647 was added after hybridization buffer heating at 75 °C. We denatured the slides at 85 °C for 5 min, and everything was performed in the dark as much as possible after TelC-Alexa647 addition.

### 2.6. Confocal Microscopy

Confocal images were acquired on a VisiScope Nikon Ti-E-based spinning disk confocal microscope (Visitron Systems; Puchheim, Germany) equipped with an iXon Ultra EMCCD camera (Andor Oxford Instruments; Abingdon, UK). All images were taken through a CFI Plan Apochromat Lambda 100× NA 1.45 objective (Nikon; Minato City, Tokyo). System control was performed by VisiView 4.3 (Visitron Systems; Puchheim, Germany).

The following dyes and emission filters were used: DAPI 460/50 nm, FITC (HHV-6 FISH) 525/50 nm, and Alexa Fluor 640 (telomere FISH) 700/75 nm. For qFISH samples, the following exposures were used: DAPI (DNA) 1.8 kW/cm 2405 nm laser, 250 ms exposure, and 500 gain; Alexa Fluor 640 (telomeres) 2.86 kW/cm 2640 nm laser, 350 ms, and 400 gain. For the dual-FISH staining, the following additional setting was used: FITC (HHV-6A) 5.7 kW/cm 2488 nm laser, 250 ms exposure, and 500 gain. All images were taken as a z-stack that was intentionally oversampled at 0.25 µm spacing intervals.

### 2.7. Microscopy Data Processing

Microscopy data were initially analyzed using the Fiji package of ImageJ 1.52, with a custom-written IJM script. All further analyses were performed with the Python programming language.

### 2.8. Statistical Analysis

Statistical analyses were performed using GraphPad Prism. The qPCR data of HHV-6A genome copies and maintenance of the integrated virus genome were analyzed using a one-way ANOVA test. Results were considered significant when *p* < 0.05.

## 3. Results

### 3.1. Telomere Length in HeLa Kyoto-Derived Cell Lines

Previous work demonstrated that telomeres carrying integrated HHV-6A/B are often the shortest in a cell [53]. However, it remained unknown whether HHV-6A/B preferentially integrate into shorter telomeres or if they shorten upon integration. To determine if telomere length influences HHV-6A integration efficiency, we used the HeLa Kyoto cell line clones that have different telomere lengths or a TZAP knockout resulting in longer telomeres [18,19]. This included (i) the parental Hela Kyoto cells; (ii) the previously published clone Hela WT Clone 5 (C5), which have been shown to display shorter telomere compared to parental cells [19]; and (iii) the Hela TZAP-KO Clone 3 (C3) and Clone 4 (C4), in which telomere length has been previously quantified using telomere restriction fragment (TRF) analyses [19,54,55,56,57].

As TRF only shows the average telomere length of a cell population, we developed a single-cell approach to investigate the uniformity of telomere length, optimizing the previously published quantitative fluorescent in situ hybridization (qFISH) method [58,59,60]. The C strand of the telomeres was stained using a PNA probe, and interphase cells imaged with a confocal microscope (Figure 1A) [61]. The intensity of all telomere puncta was extracted using ImageJ with a custom-written IJM script (see Materials and Methods) and analyzed.

The mean intensity of all telomere puncta was plotted as an empirical cumulative distribution frequency, which shows the proportion of puncta under a given fluorescence intensity (Figure 1B). Consistent with the previous TRF analysis [19], HeLa C5 had less intensely stained telomere puncta, indicating shorter telomeres (Figure 1B, pink line). In contrast, TZAP-KO C4 and the parental cells had brighter telomere puncta (Figure 1B, grey and dark blue, respectively), indicating longer telomeres. TZAP-KO C3 was included as an additional control as it has a telomere length phenotype more similar to the parental cells (Figure 1B, light blue).

To analyze the uniformity of telomere phenotypes in these cell lines, we plotted the mean total telomere intensity for each individual cell (Figure 1C). The variability between cells for each cell line is also shown in Figure 2. Parental HeLa cells had a wide range of telomere length phenotypes, while HeLa C5 had a uniform shorter telomere length phenotype for all cells in the population (Figure 1C and Figure 2B). HeLa TZAP-KO C4 displayed a telomere length variation more similar to that of parental WT cells; however, more cells displayed longer telomeres, resulting in a higher average telomere length (Figure 1C and Figure 2A vs. Figure 2D). Taken together, we used an optimized single-cell assay to assess telomere length phenotypes and confirmed the previously established cell lines that were utilized for subsequent experiments.

### 3.2. Telomere Length and HHV-6A Integration into Telomeres

To test whether host telomere length has an influence on HHV-6A integration into telomeres, we infected our panel of HeLa cells (HHV-6-negative) with defined telomere lengths with a GFP reporter HHV-6A (HHV-6A-GFP). Twenty-four hours post-infection, the number of GFP positive cells was quantified by flow cytometry and samples taken for DNA extraction. Cells were passaged for 14 days and used to assess if maintenance of the integrated virus genome was affected by the different telomere lengths by qPCR, as previously described [42]. Infection of all cell lines was comparable twenty-four hours post-infection (Figure 3, d0). At day 14, the maintenance of the viral genome was comparable between all of the cell lines, suggesting that length of the host telomeres does not influence HHV-6A integration in vitro (Figure 3, d14). Additionally, as HHV-6A could efficiently persist in the presence and absence of TZAP, this suggests that TZAP is dispensable for HHV-6A integration.

### 3.3. Length of Telomeres Harboring Integrated HHV-6

To determine if telomeres harboring the recently integrated HHV-6A are shorter, we infected 293T cells, as previously described [51,62], and assessed the length of the telomeres. Cells were infected with HHV-6A-GFP, and GFP-positive cells were sorted to obtain a pure infected population. One day post-infection, cells harboring HHV-6A were sorted, cultured until day 16, fixed, and stained for HHV-6A and telomeres by FISH. Dual staining with HHV-6 and telomere probes resulted in a robust telomere staining (Figure 4A). 

Randomly selected interphase cells containing one or multiple HHV-6A genomes were imaged and the signal of individual telomeres quantified. The colocalized signal of the telomeres and the integrated virus revealed that HHV-6A can integrate in the chromosomes of different lengths upon infection (Figure 4B). These results suggest that a general telomere shortening upon integration does not occur, as the virus could be found in short and long telomeres. However, HHV-6A appeared to integrate in many short telomeres, suggesting that, in some cases, integration may lead to a transient shortening of the telomeres, as proposed previously [25]. Taken together, our data show that telomeres harboring the integrated virus genome can substantially vary in length, while no general telomere shortening was observed.

## 4. Discussion

Here, we described the first experimental evidence that the length of host telomeres has no major effect on HHV-6A integration efficiency. Telomere length is important for the replicative capacity of a cell, where telomere length shortens with each replication until the Hayflick limit is reached and the cell either enters into senescence or dies [12,63]. Progenitor cells in early development have relatively long telomeres, a factor that is maintained in stem cells through the action of telomerase [13,14,15,16]. As herpesviruses including HHV-6A/B are maintained in the host for life, many enter latency in long-lived cells or progenitors [23,30,64]. HHV-6A/B can maintain their genome in latently infected cells by integrating into host telomeres; however, it remained unknown whether host telomere length can contribute to integration efficiency and the prevalence in certain chromosomes [25,26,29]. 

HHV-6 integration has been found in a number of telomeres, namely telomeres on 1q, 6q, 7q, 9q, 10q, 11p, 17p, 18p, 18q, 19q, 22q, and Xp [25,26,27,29,31,32,35,38,65,66]. Previous studies suggested that HHV-6A/B are often integrated in the shortest telomeres of a cell, but the reason for this has not been clearly identified [51]. One reason may be that HHV-6A/B prefer shorter telomeres or, alternatively, that the integration process shortens telomeres. The data highlighted in this study provide the first indication that both hypotheses might not be correct. Firstly, cells with different telomere lengths could maintain the HHV-6A genome to the same degree, suggesting that HHV-6A has no strong preference for telomere length during integration (Figure 3). Secondly, we showed that telomeres containing recently integrated HHV-6A can have different lengths (Figure 4B) and that integration does not shorten telomeres in general.

This last point is particularly interesting in the context of the integration mechanism. A recent study investigated the integration locus in iciHHV-6 patient samples using BioNano imaging [67,68], revealing longer-than-expected telomere sequences between the host chromosome and virus genome. These long internal telomeres would arise if integration occurs toward the telomere ends. This would in turn require neo-telomere formation on the distal end of the HHV-6A genome, contributing to the telomere signal observed in this study. 

Lastly, to assess the effect of longer host telomeres and the TZAP protein on HHV-6A integration, we utilized HeLa cells lacking a telomere-trimming molecule TZAP [19]. TZAP could possibly either impair HHV-6A integration by binding to and trimming the viral sequences or aid in the integration process by its ability to interact with the host telomeres [18]. Therefore, we utilized TZAP-KO cell lines with different telomere lengths, one longer and one similar to parental cells, in our HHV-6A integration assays (Figure 1 and Figure 2). Our data revealed that the loss of TZAP neither negatively nor positively affected HHV-6A integration (Figure 3). The involvement of cellular telomere-binding proteins such as TZAP is an exciting topic, as they may aid in HHV-6A/B integration into host telomers. A recent study revealed that the shelterin protein TRF2 not only binds the telomeric repeats in the HHV-6 genome, but is also important for efficient HHV-6 integration [69]. Further research on the role of other telomere-binding proteins should be conducted in the future to elucidate the mechanism of HHV-6 integration. 

Taken together, our data revealed that telomere length does not affect the integration efficiency of HHV-6A. In addition, integration of the virus does not induce long-term telomere shortening at least in vitro. Furthermore, the telomere-trimming molecule TZAP influences the integration efficiency of HHV-6A. These data provide important insight into the poorly understood integration mechanism of HHV-6A/B into host telomeres.

## Figures and Tables

**Figure 1 viruses-14-01864-f001:**
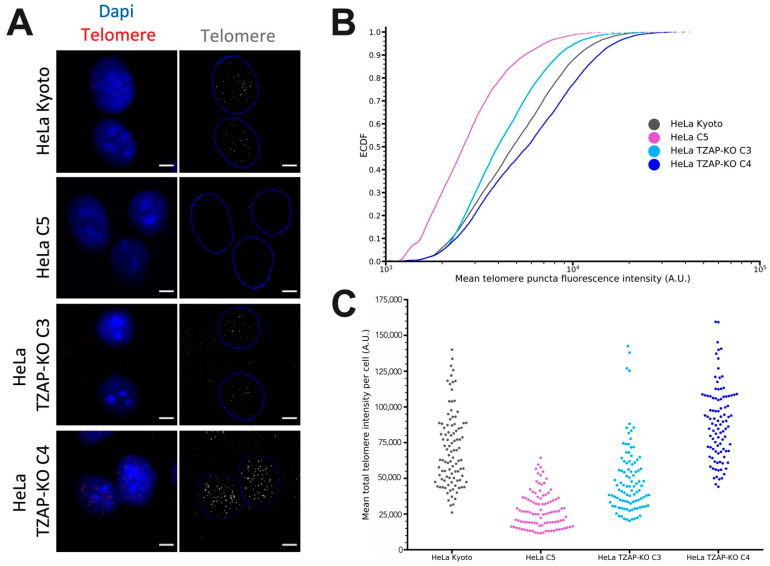
HeLa cell lines display differing telomere length phenotypes. HeLa cell lines were fixed and telomere quantitative fluorescent in situ hybridization (qFISH) was performed using a PNA probe against telomeres. (**A**) Representative images of the stained HeLa cells. DNA counterstained with DAPI (scale bar is 10 µm). (**B**) Telomere puncta intensity (mean) was measured for each indicated cell line (*n* = 100, puncta > 7500). Displayed is the cumulative distribution of puncta intensities for each cell line, where each dot represents single puncta. (**C**) For each cell, all pixels making up all puncta were summed, and an average was calculated for the cell. These values for each cell line are plotted in the swarm plot.

**Figure 2 viruses-14-01864-f002:**
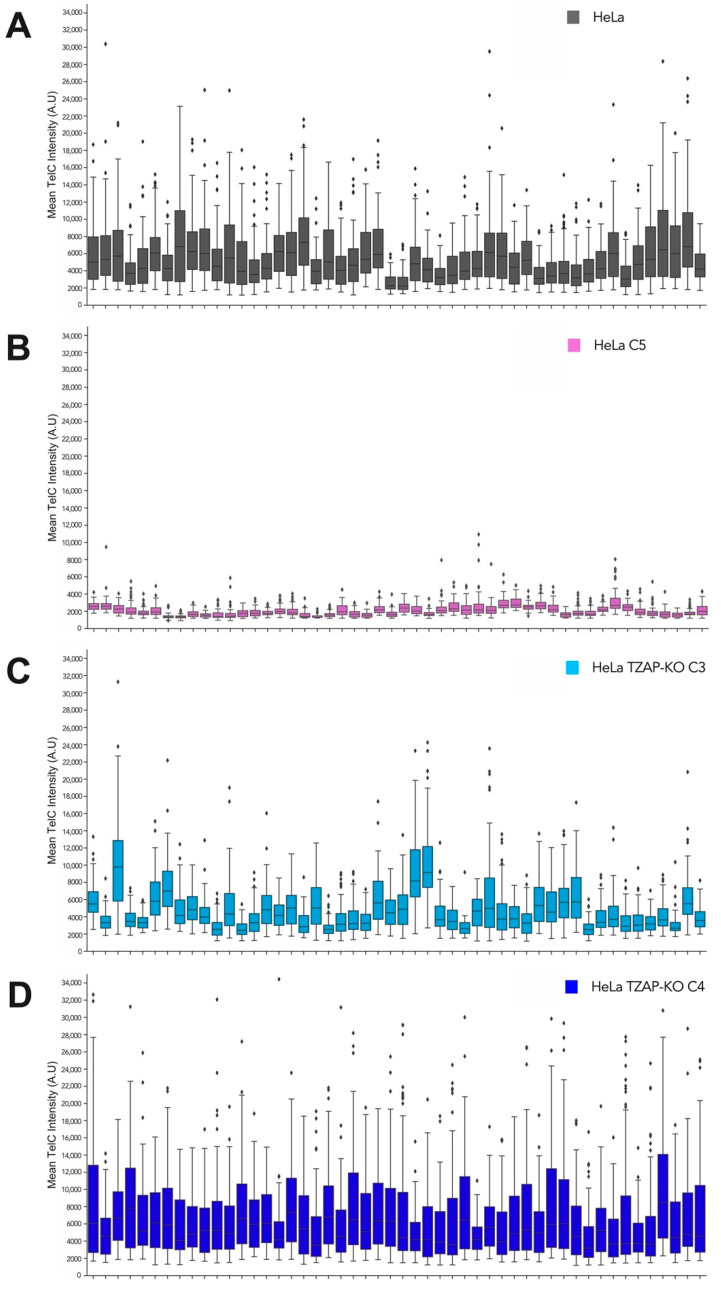
Uniformity of HeLa telomere length phenotype at individual cell level. Box plots of the mean telomere fluorescence intensity for fifty cells for (**A**) HeLa Kyoto parental, (**B**) HeLa C5, (**C**) HeLa TZAP-KO C3 and (**D**) HeLa TZAP-KO C4. Prisms are outliers defined as points > 75th quartile + 1.5* interquartile range.

**Figure 3 viruses-14-01864-f003:**
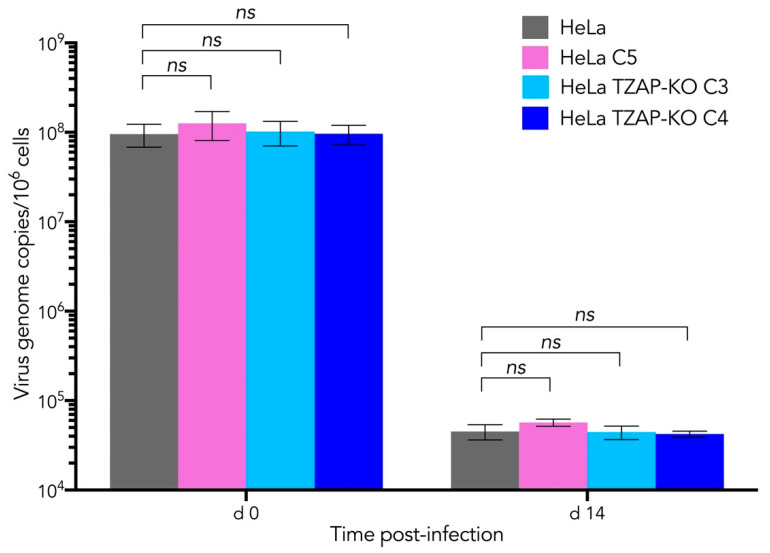
Telomere length does not influence HHV-6A integration into telomeres. The indicated HeLa cells were infected with wild-type GFP-reporter HHV-6A, and a sample was taken for DNA analysis. Fourteen days later after culturing the cells, a further DNA sample was taken. HHV-6A virus genome copies were enumerated by qPCR and normalized to cellular genome copies. Displayed are the mean number of virus genomes per million cells (*n* = 3 ± SD). No statistical significancy (ns) was detected between the different HeLa cells shortly after infection (d0) and 14 days post-infection (d14).

**Figure 4 viruses-14-01864-f004:**
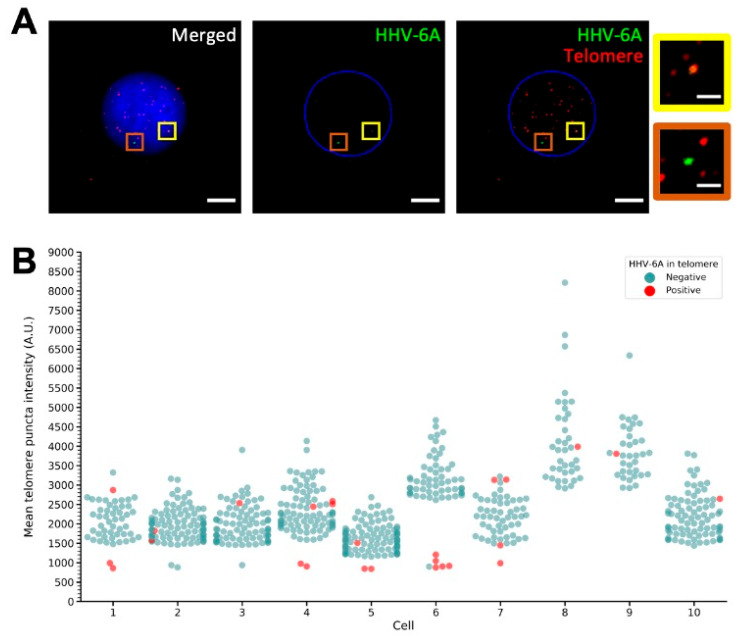
Recently integrated HHV-6A is present in telomeres with differing lengths. We infected 293T cells with wild-type GFP-reporter HHV-6A, which we sorted for GFP-positive cells, and returned to culture. On day 8, GFP expression was chemically activated and the GFP-positive cells resorted. These cells were subjected to dual FISH for the HHV-6A genome and telomeres. (**A**) Representative image of the dual FISH with the DNA counterstained with DAPI. Colored boxes mark out an HHV-6A-positive telomere containing long telomeres (yellow) and one with very short/absent telomeres (orange). Scale bars are 10 µm for large images and 2 µm for the zoomed images. (**B**) Telomere puncta intensities (mean) were measured from ten cells containing HHV-6A staining and are shown in the swarm plot (each dot is a puncta). Telomeres containing HHV-6A are shown in red and others in blue.

## Data Availability

Not applicable.

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
