# Peer review of "Impact of Host Telomere Length on HHV-6 Integration"

_viruses, 2022, doi:10.3390/v14091864_

Round 1
Reviewer 1 Report
In a published article (ref 53) Huang et al described that telomeres of cells isolated from iciHHV-6 individuals are often short in the specific chromosome end containing the integrated viral DNA using the STELA methodology. In the current manuscript Wight and colleagues describe experiments to investigate whether HHV6A integrates preferentially in cells with short telomeres of HeLa cell line-derived subclones containing telomeres of various lengths.
The authors used quantitative fluorescence in situ hybridization (FISH) to examine telomere length of various HeLa clones containing integrated HHV-6A. Overall, the data show that HHV-6A can establish latency and integrate in all available HeLa clones, regardless of their telomere length.
The authors claim that their qFISH assay is suitable for measuring telomere length. It is, however known that resolution of FISH is only 2 Megabases. Therefore, it is possible that higher FISH intensity corresponds to telomere-like sequences distal to the integrated viral genome and not genuine telomeres representing the real end of the chromosome. Currently STELA is the only method to measure the length of telomeres as used by several investigators (refs 28,29,53).
Overall, major conclusions regarding telomere length are not supported by the data because qFISH is not suitable for measuring length of pure TTAGGG repeats linked to chromosomal termini.
Author Response
We thank the reviewer for assessing our manuscript. Telomere length has been frequently assessed by qFISH in various laboratories (references 58-59-60 in the manuscript). In addition, we verified the qFISH approach on well the characterized HeLa populations and obtained data comparable to previous independent studies (reference 18 and 19 in the manuscript). To obtain the optimal resolution, images were acquired using a confocal microscope. Huang et al. indeed published an exciting study using the STELA method on iciHHV-6 patient cells, but this approach has (to our knowledge) only been successfully used on clonal iciHHV-6 cells with a known HHV-6 integration locus.
Reviewer 2 Report
HHV-6A/B viruses are unusual among herpesviruses, in that they establish latency through integration into the telomeres of chromosomes, though processes that are poorly understood. Previous studies have suggested a relationship between integration and telomere length. Here, the authors have used a novel single cell approach to investigate this question. Using cell lines with variable telomere length, they show that ability to integrate is independent of telomere length, and that integration does not shorten the telomeres. They further show that integration does not require the telomere-trimming molecule TZAP.
The studies are well designed and, on the whole, the data is convincing, although the fluorescent signal in Fig. 1A is somewhat difficult to see. This study provides novel insights into an important question pertaining to mechanisms of HHV-6 latency.
Author Response
We really appreciate the positive evaluation by the reviewer. We understand that the signal in Fig. 1A is difficult to see; however, this signal intensity is typical for FISH data. Since multiple telomere signals in the nuclei are present, a zoom in (like in Fig. 4A) could unfortunately not be used to provide a better visualization.
Reviewer 3 Report
The article by Wight et al reports that HHV6 integration into the telomeres of Hela cells does not appear to be influenced by the shortness of the telomers in a specific cell type derivative, not influenced by a mutant line in which telomeres are longer due to a factor deletion. The work also indicates that the telomere length does not appear to be strongly influenced after the integration process, in that a variety of telomere lengths can host the HHV6A genome. The work is well written, succinct and reports results of interest. I have one minor question but otherwise found no fault with the manuscript and suggest it should be published as is.
Please clarify why 293T cells were used for the work done in Figure 4? This could also do with a little detailing of telomere lengths in 293T cells and what their lengths/heterogeneity are. Are the numbers sufficient to indicate that there are more virus in the shorter lengths? I found it a little hard to understand the rationale behind the statements 210 to 216, which seem to be counteracting each other. Just needs a little rewording to be more clear
Author Response
The article by Wight et al reports that HHV6 integration into the telomeres of Hela cells does not appear to be influenced by the shortness of the telomers in a specific cell type derivative, not influenced by a mutant line in which telomeres are longer due to a factor deletion. The work also indicates that the telomere length does not appear to be strongly influenced after the integration process, in that a variety of telomere lengths can host the HHV6A genome. The work is well written, succinct and reports results of interest. I have one minor question but otherwise found no fault with the manuscript and suggest it should be published as is.
Please clarify why 293T cells were used for the work done in Figure 4? This could also do with a little detailing of telomere lengths in 293T cells and what their lengths/heterogeneity are.
Thank you for raising this point. 293T cells have been previously use for HHV-6 integration assays in several studies (references 51 and 62 in the manuscript). In addition, 293T cells have been shown to have a wide telomere length heterogeneity (Figure 4A) with generally a bit shorter telomeres than parental (normal) HeLa cells [1] [2], as also confirmed by our telomere analysis (Figure 2A and 4A)
Are the numbers sufficient to indicate that there are more virus in the shorter lengths?
Yes, we believe that this number is sufficient to show the trend that the integrated virus is often found in shorter chromosomes.
I found it a little hard to understand the rationale behind the statements 210 to 216, which seem to be counteracting each other. Just needs a little rewording to be more clear.
Thank you for picking this up. We rewrote the statement as suggested by the reviewer.
References:
- Bryan, T.M., et al., Telomere length dynamics in telomerase-positive immortal human cell populations. Exp Cell Res, 1998. 239(2): p. 370-8.
- Luo, Y., et al., Massively parallel single-molecule telomere length measurement with digital real-time PCR. Sci Adv, 2020. 6(34).